# Can ultrasound parameters predict the diagnostic yield of diuretic renal scintigraphy in pediatric hydronephrosis?

Dheeratama Siripongsatian[ID][1,2], Nipaporn Tewattanarat[3], Nantaporn Wongsurawat[1], Suwannee Wisanuyotin[ID][4], Daris Theerakulpisut[ID][1]*

1 Division of Nuclear Medicine, Department of Radiology, Faculty of Medicine, Khon Kaen University, Khon Kean, Thailand, 2 National Cyclotron and PET Centre, Chulabhorn Hospital, Bangkok, Thailand, 3 Division of Diagnostic Radiology, Department of Radiology, Faculty of Medicine, Khon Kaen University, Khon Kean, Thailand, 4 Division of Pediatric Nephrology, Department of Pediatrics, Faculty of Medicine, Khon Kaen University, Khon Kaen, Thailand

* daristh@kku.ac.th

## Abstract

Diuretic renal scintigraphy (DRS) is a valuable imaging tool for distinguishing obstructive from non-obstructive hydronephrosis, especially in pediatric patients, but non-diagnostic results are not uncommon. In this retrospective analytical study, we examined if kidney ultrasound (US) parameters can predict non-diagnostic DRS, and DRS with an obstruction pattern, potentially guiding patient selection to reduce unnecessary radiation exposure. The study included 67 patients (134 kidneys), 1-month to 4-years of age who underwent both DRS and US. Receiver operating characteristic (ROC) curve analysis of US parameters including parenchymal thickness, cortical thickness, medullary pyramidal thickness, anterior-posterior renal pelvic diameter, was done to assess the predictiveness of these parameters for prediction of DRS results. None of the US parameters reliably predicted non-diagnostic DRS results (AUC range: 0.41–0.61). However, these parameters demonstrated good predictiveness for identifying DRS with an obstruction pattern (AUC range: 0.69–0.87), with anterior-posterior renal pelvic diameter showing the highest performance. These findings suggest that while US parameters cannot predict non-diagnostic DRS outcomes, they are effective in identifying obstruction patterns on DRS.

## Introduction

Hydronephrosis, a condition characterized by the dilatation of the renal pelvis and calyces, can occur in patients of all ages. In the pediatric population, this condition may be congenital and has been reported in 0.6–4.5% of pregnancies [1,2]. Most cases of fetal hydronephrosis are attributed to transient physiological changes during early development, such as temporary narrowing at the ureteropelvic junction which generally resolve spontaneously [3]. However, up to one-third of infants with

**Data availability statement:** All relevant data are within the paper and its Supporting information files.

**Funding:** The author(s) received no specific funding for this work.

**Competing interests:** The authors have declared that no competing interests exist.

hydronephrosis are found to have congenital anomalies of the kidney and urinary tract [4]. These structural abnormalities can impair normal kidney development and function, potentially progressing to chronic kidney disease or renal failure if not appropriately diagnosed and managed [5].

Differentiating patients with obstructive hydronephrosis from those with non-obstructive hydronephrosis is crucial in planning appropriate patient management, since the former often requires surgical interventions to prevent deterioration of renal function [5,6]. Although ultrasound (US) is the imaging of choice in evaluating children with hydronephrosis and is the initial imaging recommendation in most clinical scenarios [7], it cannot distinguish between obstructive and non-obstructive hydronephrosis.

Diuretic renal scintigraphy (DRS), a nuclear medicine imaging technique, can differentiate between obstructive and non-obstructive hydronephrosis. Those with non-obstructive causes will show remarkable improvement in radiotracer excretion after diuretic administration, but those with obstructive causes will show continued retention of the radiotracer in the kidney [8]. However, several factors can interfere with the performance of DRS which may result in an inconclusive scan, which not only fails to provide actionable clinical information but also subjects the patient to unnecessary radiation exposure. Among the most common challenges are cases of hydronephrosis with excessive renal pelvic volume, where the large 'reservoir effect' prevents adequate clearance despite the diuretic-induced increased urine flow. Another common cause is severely impaired function of the hydronephrotic kidney, which can delay or diminish the diuretic response. The degree of hydronephrosis can be directly measured by US. The parenchymal thickness may be a surrogate representation of renal function, because as hydronephrosis progresses, the kidney function will deteriorate as the kidney parenchyma becomes progressively thinner. For these reasons, we hypothesize that US parameters should be able to predict if DRS will be diagnostic or non-diagnostic, and could help to better select patients who would most likely benefit from DRS. The primary objective of this study is to investigate the ability of US to predict diagnostic vs. non-diagnostic DRS, and the secondary objective is to determine its ability to predict obstruction vs. other categories of DRS findings.

## Materials and methods

### Patients

This single-center, retrospective analytical predictive study was approved by the Khon Kaen University Ethics Committee for Human Research (protocol number. HE611197) on May 8th, 2018. Due to its retrospective nature, requirements for obtaining informed consent were waived by the ethics committee. Access to patient data began approximately in June 2018, after ethics committee approval was granted. The authors had access to patients' identifying information during the data collection process. Only patient hospital numbers are kept in the final dataset, used only for verification purposes, and is kept strictly confidential. No other identifying information is maintained. The dataset in the Supporting Information is completely anonymized and cannot be used to identify any of the patients.

Included were pediatric patients 1 month to 4 years of age who underwent both DRS and kidney US at the Department of Radiology, Faculty of Medicine, Khon Kaen University from January 1st, 2013, to August 31st, 2019. Excluded were patients whose kidney US and DRS were done more than 6 months apart and those with uninterpretable DRS due to technical factors such as significant patient motion.

## Kidney ultrasound

Ultrasound images were retrieved from the picture archiving and communication system of our hospital. A diagnostic radiologist specializing in pediatric imaging reviewed the images and measured key parameters of interest. Measurements were made on sagittal plane images. The cortical thickness (CT), medullary pyramid thickness (MT), and parenchymal thickness (PT) were measured perpendicular to the renal capsule. The CT is the shortest distance from the base of the medullary pyramid to the renal capsule, the MT is the distance from the base to apex of the medullary pyramid, the PT is the distance from the renal capsule to the apex of the medullary pyramid. The kidney size (KS) is the distance from the upper to lower pole of the kidney. The anterior-posterior renal pelvic diameter (apRPD) was measured as the distance of maximum distension of renal pelvis from the innermost part of the intrarenal pelvis to the outermost part of the extrarenal pelvis. Fig 1 depicts measurement for each parameter. The Society of Fetal Urology (SFU) grading of hydronephrosis [9] was also documented.

## Diuretic renal scintigraphy technical procedure

Diuretic renal scintigraphy was performed according to a standard institutional protocol. To ensure proper hydration, patients were encouraged to drink milk or water before commencing the study. An indwelling IV catheter attached to a three-way stopcock was inserted into a superficial vein at either the upper or lower extremity. Image acquisition was done with the patient in supine position under a Discovery NM/CT 670 gamma camera and SPECT/CT system (GE Healthcare, IL, USA) equipped with a low-energy all-purpose collimator. Posterior dynamic imaging using list mode acquisition was started simultaneously with injection of 20 MBq of [99mTc] Tc-MAG3, using a 1,024 × 1,024 × 16 matrix, photopeak of 140 keV ± 15%. Images were acquired in two phases. The first phase using a frame rate of 1 seconds/frame during the first 60 seconds, and the second phase using a frame rate of 20 seconds/frame for the remaining 30 minutes. A 1-mg/kg dose of furosemide was administered intravenously midway into the image acquisition at the 15-minute mark. Radioactivity in the syringe was measured before and after injection. All studies were reanalyzed by D.S. (2 years or experience) under supervision of a nuclear medicine physician, D.T. (9 years of experience), using Xeleris 3.0 software provided by the

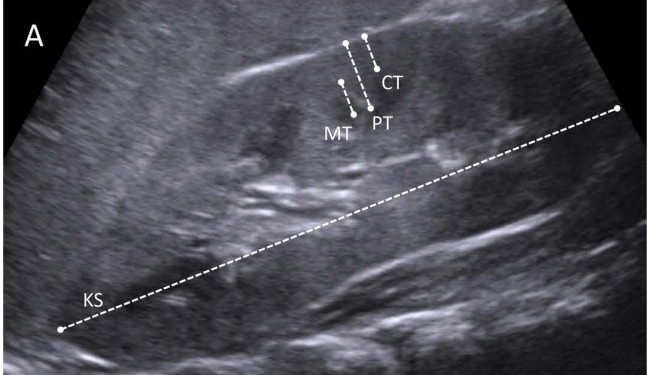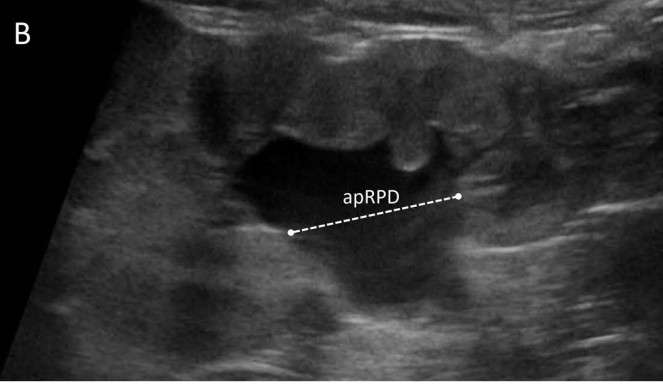

**Fig 1. Measurement of ultrasound parameters.** A: measurement of kidney size (KS) as represented by kidney length, parenchymal thickness (PT), cortical thickness (CT), and medullary pyramidal thickness (MT). B: measurement of anterior-posterior renal pelvic diameter (apRPD).

manufacturer. Whole kidney regions of interest (ROI) were drawn around both kidneys, semilunar background ROIs were places inferolaterally to each kidney.

### Diuretic renal scintigraphy interpretation

Interpretation of DRS is based on a constellation of findings from the scintigraphic images, as well as the renogram, and quantitative parameters [8]. For this study, kidneys were categorized into 4 groups: 1) Normal, when there is spontaneous excretion of the radiotracer even before diuretic administration and no significant retention in the kidney at the end of the study; 2) Dilated without obstruction, when there is retention of the radiotracer in the kidney which promptly cleared out after furosemide administration and no significant retention in the kidney at the end of the study; 3) Obstruction, when there is retention of the radiotracer in the kidney with no or minimal response to furosemide and significant retention in the kidney at the end of the study; and 4) Indeterminate, when the kidney cannot be placed into either of the three preceding categories, for example, kidneys with poor function with no or minimal radiotracer accumulation, markedly dilated kidneys with minimal response to furosemide which could be due to a reservoir effect. The diuretic excretion half-time ($T_{1/2}$), i.e., the time it takes until the activity in the kidney falls from its peak to 50% of its peak in response to diuretic, was used as a supporting parameter. The $T_{1/2}$ for 'dilated without obstruction', 'obstruction', and 'indeterminate', is often < 10 minutes, 10–20 minutes, and > 20 minutes, respectively. The $T_{1/2}$ was never used alone in making the classification. Fig 2 provides examples of DRS findings representative of each category. Among the 4 categories, 'normal', 'dilated without obstruction', and 'obstruction' were considered as "diagnostic" whereas 'indeterminate' was considered "non-diagnostic".

### Statistical analysis

Patient demographics and imaging findings are presented using descriptive statistics. Continuous data were summarized as mean and standard deviation. Categorical data are expressed as counts and percentages. To investigate the ability of US parameters to predict if a DRS would be "diagnostic" or "non-diagnostic", receiver operating characteristic (ROC) curve analysis and the area under the ROC curve (AUC) were determined for each US parameter, using the logistic regression framework. Accompanying 95% confidence intervals were reported as appropriate. Statistical analysis was done using Stata version 18.5 (StataCorp LLC., College Station, TX)

### Results

From January 1st, 2013, to August 31st, 2019, a total of 76 patients underwent DRS and US, 9 cases had significant motion artifacts in the DRS and were excluded, leaving a total of 67 patients (134 kidneys) in the final analysis. Most of the cohort were males, with an average age of 13.6 ± 12.5 months. Patient characteristics are summarized in Table 1. All patients had good overall renal function as indicated by the average serum creatinine of 0.3 ± 0.1 mg/dL.

Ultrasound findings are summarized in Table 2. Approximately 30% of the kidneys were normal without dilatation with SFU grade 0, approximately 40% had hydronephrosis with SFU grades 1–3, and approximately 20% had severe hydronephrosis with parenchymal compromise with SFU grade 4.

Diuretic renal scintigraphy findings are given in Table 3, 25.4% of kidneys had normal DRS, 25.4% were dilated without obstruction, and 26.1% were obstructed, totaling 76.9% of kidneys classified as having a "diagnostic" DRS whereas the remaining 23.1% was indeterminate and classified as "non-diagnostic".

Ultrasound parameters according to the four categories of diuretic renal scintigraphy results are presented in Table 4 with the accompanying boxplots shown in Fig 3. For thickness parameters normal DRS has the most thickness followed by dilated without obstruction, indeterminate, with obstruction having the lowest thickness. The renal pelvic diameter was notably higher in the obstruction group compared with the remaining three categories which had similar values. The kidney size was also found to be highest in the obstruction group, but the difference compared with the remaining three groups was not as pronounced.

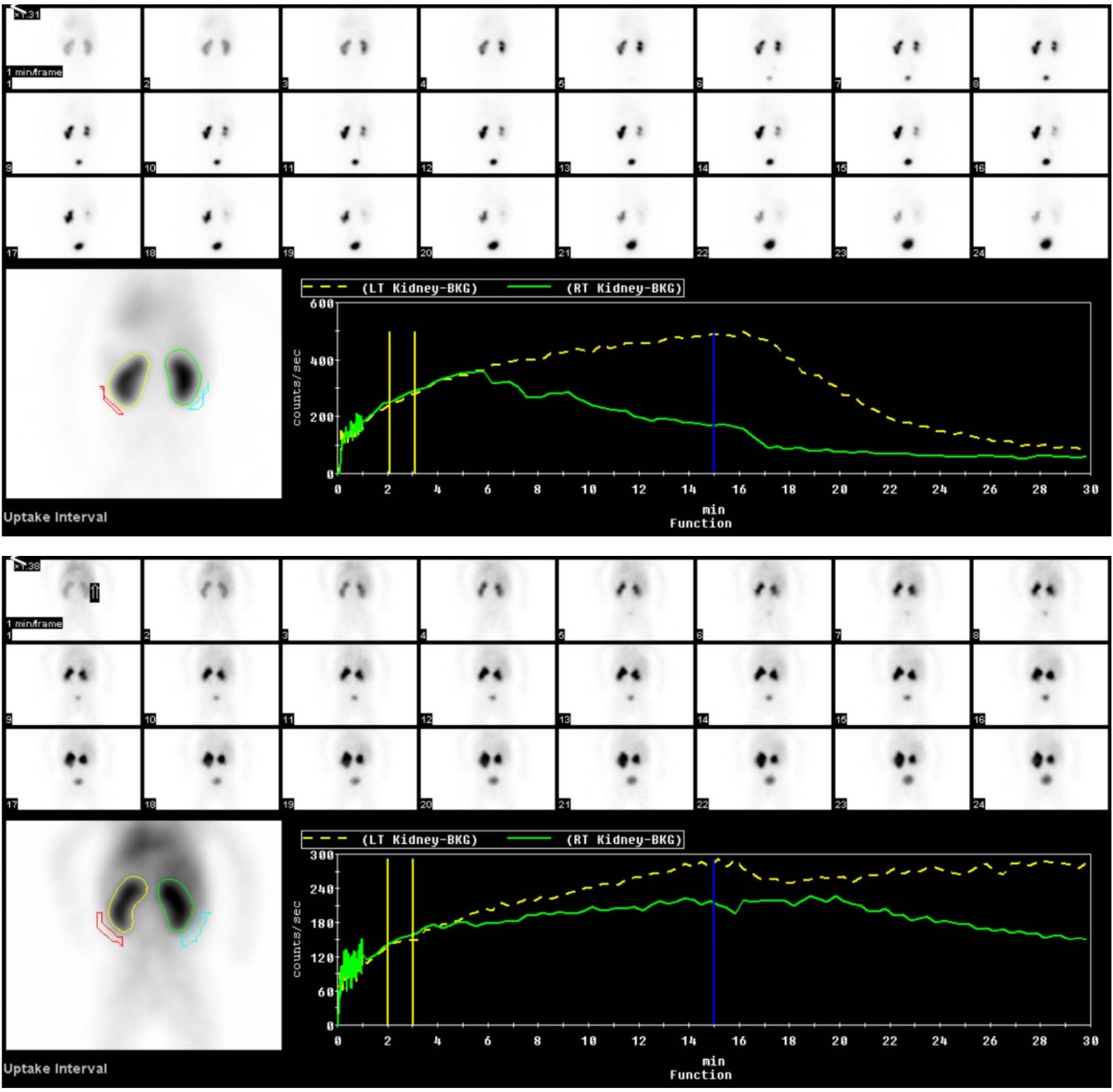

**Fig 2. Examples of diuretic renal scintigraphy results.** Top: A diuretic renal scintigraphy of a 2-year-old boy with left hydronephrosis showing radio-tracer retention with prompt improvement in excretion after furosemide administration consistent with dilatation without obstruction, and normal excretion of the normal right kidney. Bottom: A diuretic renal scintigraphy of an 8-month-old girl with bilateral hydronephrosis showing marked radiotracer retention in both kidneys. The left kidney shows no improvement after furosemide administration which is consistent with obstruction, whereas the right kidney shows some improvement of excretion but still with significant retention at the end of the study which is considered an indeterminate result.

**Table 1. Patient characteristics.**

| Characteristics | |
|---|---|
| Number | 67 (100%) |
| Sex (count, %) | |
| Male | 54 (80.6%) |
| Female | 13 (19.4%) |
| Age (months; mean ± SD) | 13.6 ± 12.5 |
| Age range (count, %) | |
| 1 to ≤ 4 months | 21 (31.3%) |
| 4 to ≤ 8 months | 11 (16.4%) |
| 8 to ≤ 12 months | 7 (10.5%) |
| 12 to ≤ 24 months | 14 (20.9%) |
| 24 to ≤ 36 months | 9 (13.4%) |
| > 36 months | 5 (7.5%) |
| Body weight (kilograms; mean ± SD) | 8.9 ± 3.4 |
| Height (centimeters; mean ± SD) | 70.9 ± 13.9 |
| Serum creatinine (mg/dL) | |
| Mean ± SD | 0.3 ± 0.1 |
| Range | 0.2–0.8 |

A summary of patient characteristics. Continuous variables are expressed as mean values ± standard deviations (SD). Categorical variables are expressed as counts and percentage.

**Table 2. Ultrasound findings.**

| Parameter | |
|---|---|
| Anterior-posterior renal pelvic diameter (mm; mean ± SD) | 8.4 ± 9.6 |
| Parenchymal thickness (mm; mean ± SD) | 8.5 ± 3.6 |
| Cortical thickness (mm; mean ± SD) | 3.3 ± 1.5 |
| Medullary pyramid thickness (mm; mean ± SD) | 5.2 ± 2.6 |
| Kidney size (mm; mean ± SD) | 60.9 ± 13.6 |
| SFU grading (count, %) | |
| Grade 0 | 42 (31.3%) |
| Grade 1 | 28 (20.9%) |
| Grade 2 | 21 (15.7%) |
| Grade 3 | 10 (7.5%) |
| Grade 4 | 27 (20.1%) |
| Unevaluable | 6 (4.5%) |

A summary of ultrasound finding among the 134 kidneys (of 67 patients) examined. Continuous variables are expressed as mean values ± standard deviations (SD). Categorical variables are expressed as counts and percentage. mm, millimeter; SFU, Society of Fetal Urology.

Results of receiver operating characteristic curve analysis of ultrasound parameters for predicting DRS results are presented in Table 5 and their ROC curves are presented in Fig 4. For distinguishing between those with diagnostic and non-diagnostic DRS, all US parameters had poor discriminatory power with AUCs ranging from 0.41–0.61 with all values having 95% confidence intervals crossing 0.5 which indicates that none of the US parameters can be used to predict if the

**Table 3. Diuretic renal scintigraphy findings.**

| Parameter | |
|---|---|
| Effective renal plasma flow (mL/min, mean±SD) | 162.9±67.4 |
| Interpretation (count, %) | |
| Diagnostic | |
| Normal | 34 (25.4%) |
| Dilated without obstruction | 34 (25.4%) |
| Obstruction | 35 (26.1%) |
| Non-diagnostic | |
| Indeterminate | 31 (23.1%) |

A summary of diuretic renal scintigraphy finding among the 134 kidneys (of 67 patients) examined. Continuous variables are expressed as mean values±standard deviations (SD). Categorical variables are expressed as counts and percentage. mL/min, milliliters per minute.

**Table 4. Ultrasound parameters according to diuretic renal scintigraphy results.**

| Parameter | Normal | Dilated without obstruction | Indeterminate | Obstruction |
|---|---|---|---|---|
| Parenchymal thickness (mm) | 10.8±2.1 | 9.7±2.5 | 7.1±4.2 | 6.3±3.3 |
| Cortical thickness (mm) | 4.0±1.4 | 3.7±1.4 | 2.8±1.4 | 2.6±1.2 |
| Medullary pyramidal thickness (mm) | 6.8±1.5 | 5.9±1.8 | 4.3±3.1 | 3.6±2.7 |
| Anterior-posterior renal pelvic diameter (mm) | 2.2±2.7 | 5.8±5.0 | 8.6±9.6 | 18.1±11.0 |
| Kidney size (mm) | 57.6±9.2 | 57.6±12.6 | 58.0±14.1 | 70.2±14.0 |

Summary of ultrasound parameters across the four categories of diuretic renal scintigraphy results. Units are in millimeters (mm). Data are presented as mean and standard deviation (SD).

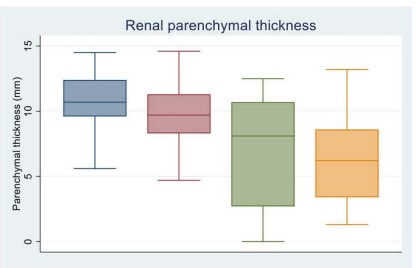
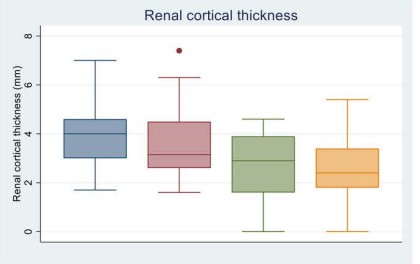
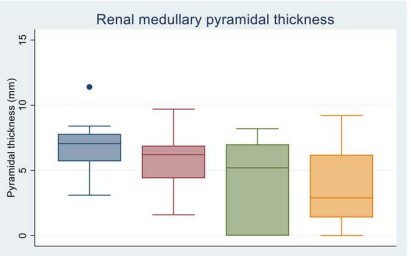
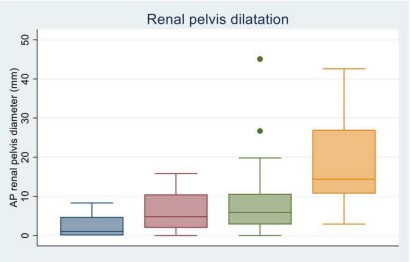
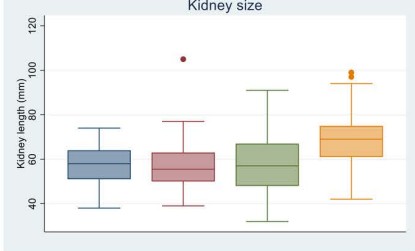
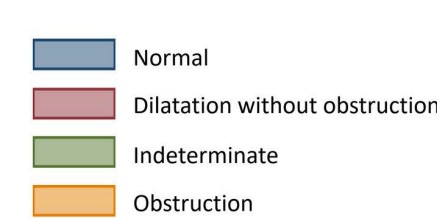

**Fig 3. Boxplots of ultrasound parameters.** Parameters including renal parenchymal thickness, renal cortical thickness, renal medullary pyramidal thickness, anterior-posterior renal pelvic diameter, and kidney size, summarized across four categories of diuretic renal scintigraphy results.

**Table 5. Results of receiver operating characteristic curve analysis.**

| Ultrasound parameter | Diagnostic vs. non-diagnostic | | Obstruction vs. other categories | |
|---|---|---|---|---|
| | AUC | 95% CI. | AUC | 95% CI. |
| Parenchymal thickness | 0.61 | 0.49 - 0.74 | 0.74 | 0.64 - 0.84 |
| Cortical thickness | 0.60 | 0.49 - 0.72 | 0.69 | 0.59 - 0.79 |
| Medullary pyramid thickness | 0.59 | 0.46 - 0.72 | 0.73 | 0.62 - 0.83 |
| Kidney length | 0.41 | 0.29 - 0.54 | 0.79 | 0.69 - 0.88 |
| Anterior-posterior renal pelvic diameter | 0.53 | 0.42 - 0.65 | 0.87 | 0.80 - 0.93 |

Results of receiver operating characteristic curve analysis of ultrasound parameters for predicting diagnostic vs. non-diagnostic diuretic renal scintigraphy results, and obstruction vs. all other categories. AUC, area under receiver operating characteristic curve; CI., confidence interval.

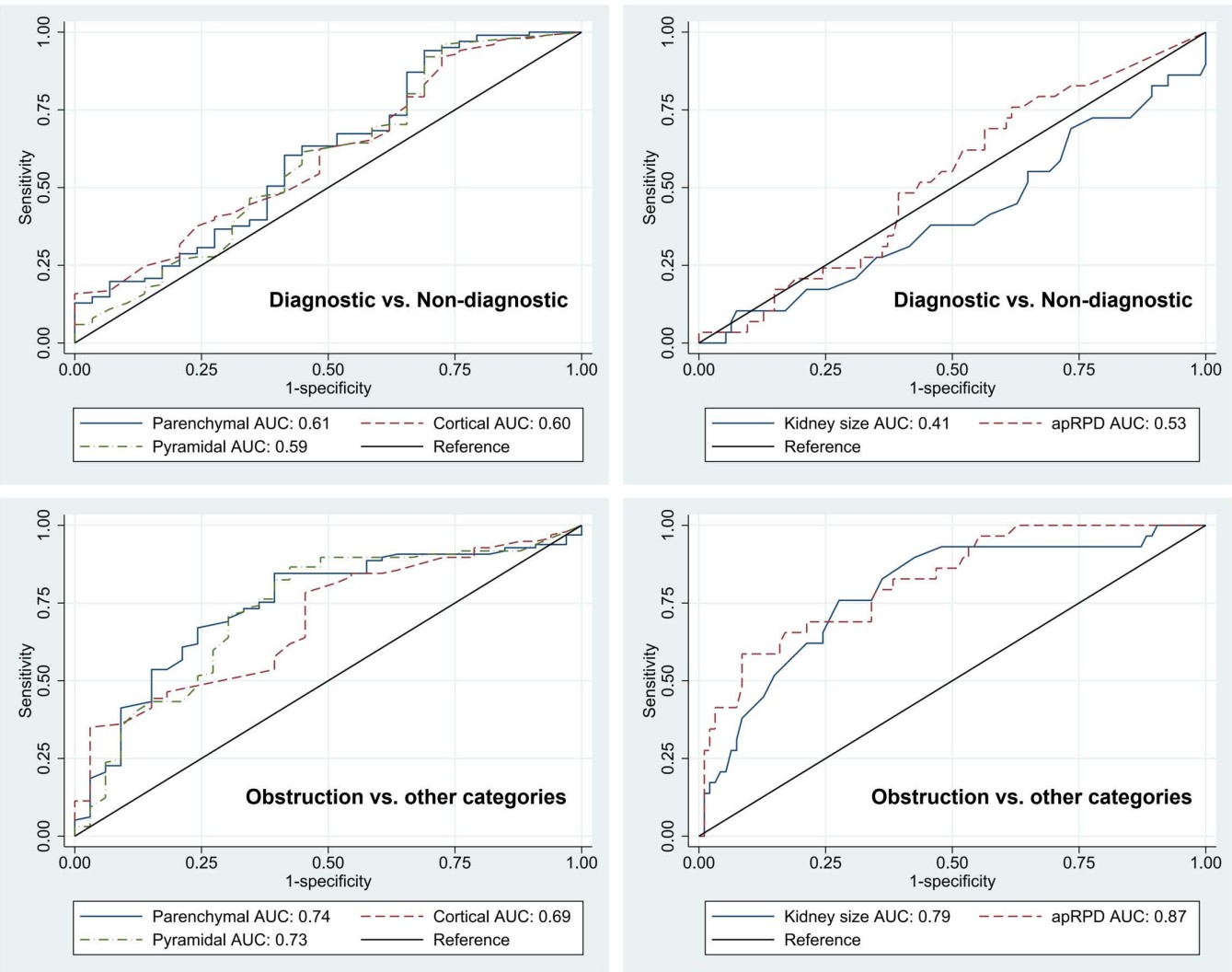

**Fig 4. Receiver operating characteristic curves (ROC) of ultrasound parameters with accompanying area under the curve (AUC).** Top left: ROCs of parenchymal thickness, cortical thickness, and medullary pyramidal thickness for distinguishing diagnostic vs. non-diagnostic DRS. Top right: ROCs of kidney size and anterior-posterior renal pelvic diameter (apRPD) for distinguishing diagnostic vs. non-diagnostic DRS. Bottom left: ROCs of parenchymal thickness, cortical thickness, and medullary pyramidal thickness for distinguishing obstruction vs. other categories of DRS. Bottom right: ROCs of kidney size and apRPD for distinguishing obstruction vs. other categories of DRS.

**Table 6. Results of receiver operating characteristic curve analysis (subgroup analysis of cases where the time window between the diuretic renal scintigraphy and ultrasound is ≤ 90 days).**

| Ultrasound parameter | Diagnostic vs. non-diagnostic | | Obstruction vs. other categories | |
|---|---|---|---|---|
| | AUC | 95% CI. | AUC | 95% CI. |
| Parenchymal thickness | 0.54 | 0.38 - 0.70 | 0.81 | 0.71 - 0.91 |
| Cortical thickness | 0.49 | 0.33 - 0.64 | 0.79 | 0.67 - 0.90 |
| Medullary pyramid thickness | 0.56 | 0.40 - 0.71 | 0.79 | 0.67 - 0.91 |
| Kidney length | 0.54 | 0.40 - 0.69 | 0.80 | 0.66 - 0.93 |
| Anterior-posterior renal pelvic diameter | 0.46 | 0.32 - 0.60 | 0.88 | 0.79 - 0.97 |

Results of receiver operating characteristic curve analysis of ultrasound parameters for predicting diagnostic vs. non-diagnostic diuretic renal scintigraphy results, and obstruction vs. all other categories; a subgroup analysis of cases where the time window between the diuretic renal scintigraphy and ultrasound is ≤ 90 days. AUC, area under receiver operating characteristic curve; CI., confidence interval.

DRS would be diagnostic or non-diagnostic. Conversely, the US parameters had good predictiveness for distinguishing DRS with obstruction from the remaining three categories, with AUCs ranging from 0.69–0.87.

The median time window between DRS and US was 44 days. To further examine if the time window between DRS and US could have affected the predictiveness of US parameters, a subgroup analysis was done in 44 of 67 patients (65.7%) who underwent both modalities within ≤ 90 days of each other. The results are presented in Table 6. The AUCs of the US parameters did not improve, ranging from 0.46–0.56, with all having 95% confidence intervals crossing 0.5. This suggests that the time window is not an effect modifier on the predictiveness of US parameters.

## Discussion

Diuretic renal scintigraphy (DRS) is a safe and valuable nuclear medicine imaging technique widely used to differentiate obstructive from non-obstructive causes of hydronephrosis in pediatric patients, but DRS can often have inconclusive results. In this study, we investigated whether US parameters can predict if DRS results would be diagnostic or non-diagnostic, which could help better select patients who would benefit from DRS and avoid unnecessary radiation exposure in those likely to have non-diagnostic DRS.

Among the most common challenges of DRS interpretation are cases of hydronephrosis with excessive renal pelvic volume, where the large 'reservoir effect' prevents adequate clearance despite the diuretic-induced increased urine flow, and severely impaired function of the hydronephrotic kidney, which can delay or diminish the expected diuretic response. We believed that both of these factors can be estimated by US parameters, with kidney size and apRPD representing the hydronephrotic volume, and the thickness parameters as surrogates for renal function. However, as the results show, all the US parameters studied failed to distinguish between diagnostic vs. non-diagnostic DRS. For the thickness parameter, the average values of DRS with indeterminate (non-diagnostic) results fall between the average values of the normal, dilated without obstruction, and the obstruction (diagnostic) groups. This results in the overall values of the diagnostic and non-diagnostic DRS groups being overall similar thus explaining their poor discriminating power. For the kidney size and the apRPD, these parameters were also not predictive which could be because these parameters were similar among the normal, dilatation without obstruction, and the indeterminate groups. These findings indicate that there is no single simple US parameter that could predict a non-diagnostic DRS since there is no single cause of a non-diagnostic scan, but rather a composite of multiple factors including but not limited to the severity of hydronephrosis and kidney function. A previous study by Jacobson et al. found a poor correlation between DRS and US findings in children with hydronephrosis with a Spearman correlation coefficient of only 0.24 [10]. However, to our knowledge, this current study is the first to examine the predictiveness of US parameters in relation to diagnostic vs. non-diagnostic DRS results.

On the other hand, all US parameters performed rather well in predicting DRS with obstruction, with apRPD having the best predictiveness with an AUC of 0.87. This is consistent with findings from previous studies. Rianthavorn and Limwattana found that apRPD measured by US had an AUC of 0.86 in predicting obstructive hydronephrosis [11]. Błaszczyk et al. reported an apRPD cutoff of ≥ 23mm had a sensitivity of 94%, specificity of 76%, accuracy of 80%, and AUC of 0.91 for prediction of obstructive hydronephrosis [12]. Other studies also found that apRPD was able to distinguishing obstructive from non-obstructive cases although AUCs were not reported [13–15].

The main limitation of this study is its retrospective nature. Although all US studies were done by pediatric radiologists, since US is a highly operator-dependent modality, variations at the time the US studies were done are difficult to avoid. To address this issue, in this study, all US parameters were remeasured by one experienced pediatric radiologist (N.T.). Future prospective studies should aim to integrate multiple clinical, functional, and anatomical parameters to enhance prediction accuracy and improve patient selection for DRS.

## Conclusion

While ultrasound remains an essential tool for initial assessment, its parameters alone may not reliably predict the diagnostic yield of DRS. Future studies should aim to integrate multiple clinical, functional, and anatomical parameters to enhance prediction accuracy and improve patient selection for DRS.

## Supporting information

**S1 File.** Anonymized patient-level dataset.
(CSV)

## Acknowledgments

Not applicable.

## Author contributions

**Conceptualization:** Dheeratama Siripongsatian, Daris Theerakulpisut.

**Data curation:** Dheeratama Siripongsatian.

**Formal analysis:** Daris Theerakulpisut.

**Investigation:** Dheeratama Siripongsatian, Nipaporn Tewattanarat, Nantaporn Wongsurawat, Suwannee Wisanuyotin, Daris Theerakulpisut.

**Methodology:** Daris Theerakulpisut.

**Project administration:** Daris Theerakulpisut.

**Supervision:** Daris Theerakulpisut.

**Validation:** Suwannee Wisanuyotin, Daris Theerakulpisut.

**Visualization:** Dheeratama Siripongsatian, Daris Theerakulpisut.

**Writing – original draft:** Dheeratama Siripongsatian, Daris Theerakulpisut.

**Writing – review & editing:** Dheeratama Siripongsatian, Nipaporn Tewattanarat, Nantaporn Wongsurawat, Suwannee Wisanuyotin, Daris Theerakulpisut.

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
