## [Decision Letter · Decision Letter 0]

14 Mar 2025

PONE-D-25-01165Can ultrasound parameters predict the diagnostic yield of diuretic renal scintigraphy in pediatric hydronephrosis?PLOS ONE

Dear Dr. Theerakulpisut,

Thank you for submitting your manuscript to PLOS ONE. After careful consideration, we feel that it has merit but does not fully meet PLOS ONE’s publication criteria as it currently stands. Therefore, we invite you to submit a revised version of the manuscript that addresses the points raised during the review process.

We look forward to receiving your revised manuscript.

Kind regards,

Ahmet Çağlar, M.D.

Academic Editor

PLOS ONE

2. Please include captions for your Supporting Information files at the end of your manuscript, and update any in-text citations to match accordingly. Please see our Supporting Information guidelines for more information: http://journals.plos.org/plosone/s/supporting-information .

Reviewers' comments:

Reviewer's Responses to Questions

**Comments to the Author**

1. Is the manuscript technically sound, and do the data support the conclusions?

Reviewer #1: Yes

Reviewer #2: Yes

2. Has the statistical analysis been performed appropriately and rigorously? 

Reviewer #1: Yes

Reviewer #2: Yes

3. Have the authors made all data underlying the findings in their manuscript fully available?

Reviewer #1: Yes

Reviewer #2: Yes

4. Is the manuscript presented in an intelligible fashion and written in standard English?

Reviewer #1: Yes

Reviewer #2: Yes

5. Review Comments to the Author

Reviewer #1: The manuscript technically sound, and do the data support the conclusions.It has the statistical analysis been performed appropriately and rigorously. The authors made all data underlying the findings in their manuscript fully available.the manuscript presented in an intelligible fashion and written in standard English

Reviewer #2: In the current work, authors performed a retrospective study to study the ability of kidney ultrasound and its paramters to distinguish between diagnostic and non-diagnostic DRS so that people who dont qualify for diagnostic DRS can avoid DRS. The study found that kidney ultasound canot predict DRS with significance. The manuscript is well written and the data supports the conclusions. Statistical analysis is well done. There are a few minor comments if authors can address.

1) Can some anotomical changes occur in the 6 month time gap between kidney US and DRS? What if the exclusion criteria is more tight (3 months or lower)? Will that make any difference in the results?

2) are there any previous studies that have similarly explored the predictive ability of Kidney US? The authors have not mentioed about them. It would be intersting to know what is the outcome in the previous studies.

3) Authors can mention any limiations in the current study. Also, mention the future directions for the researchers reading the manuscript.

6. PLOS authors have the option to publish the peer review history of their article (what does this mean? ). If published, this will include your full peer review and any attached files.

**Do you want your identity to be public for this peer review?** For information about this choice, including consent withdrawal, please see our Privacy Policy .

Reviewer #1: No

Reviewer #2: No

---

## [Author Response · Author response to Decision Letter 0]

30 Mar 2025

### Comments from the Editorial office:

#Comment:

# Response:

File names are renamed to comply with PLOS ONE's style requirements. (Fig1.tif, Fig2.tif, Fig3.tif, Fig4.tif, S1_Dataset.csv)

# Comment: 2. Please include captions for your Supporting Information files at the end of your manuscript, and update any in-text citations to match accordingly.

# Response: A caption for the Supporting Information file is included at the end of the revised manuscript.

# Revised text:

"Supporting information

S1 Dataset. Anonymized patient-level dataset"

# Comment: 3. Please review your reference list to ensure that it is complete and correct. If you have cited papers that have been retracted, please include the rationale for doing so in the manuscript text, or remove these references and replace them with relevant current references. Any changes to the reference list should be mentioned in the rebuttal letter that accompanies your revised manuscript. If you need to cite a retracted article, indicate the article’s retracted status in the References list and also include a citation and full reference for the retraction notice.

# Response:

• The references were checked and revised to comply with the Vancouver style.

• An additional reference is added as per recommendation from the reviewer (reference number 10) with the references that follow renumbered accordingly.

# Revised text:

10. Jacobson DL, Flink CC, Johnson EK, Maizels M, Yerkes EB, Lindgren BW, et al. The correlation between serial ultrasound and diuretic renography in children with severe unilateral hydronephrosis. J Urol. 2018 Aug;200(2):440–7.

### Comments from Reviewer 2:

# Comment: 1) Can some anotomical changes occur in the 6 month time gap between kidney US and DRS? What if the exclusion criteria is more tight (3 months or lower)? Will that make any difference in the results?

# Response: We agree that this is an important point, and it is one that we have considered when initiating the study. We believe that anatomical changes that could occur is possible, but should not be so pronounced as to invalidate the results of the study. This is because according to the “Pediatric congenital hydronephrosis (ureteropelvic junction obstruction): Medical management guide” (reference 5), children with hydronephrosis are initially managed medically with serial ultrasound follow-up every 3-6 months, which suggests that significant changes in the kidney anatomy would not occur in a short time frame. However, to explore if including only those that underwent DRS and US within 3 months (90 days) of each other, we did a subgroup analysis to find the ROCs of US parameters in 44 patients whose imaging tests were done within a 3-month window. The results are presented in a new table (Table 6). The AUCs are still poor, so we believe that the time window did not have a significant effect on the outcome and the key message of this study.

# Revised text:

"The median time window between DRS and US was 44 days. To further examine if the time window between DRS and US could have affected the predictiveness of US parameters, a subgroup analysis was done in 44 of 67 patients (65.7%) who underwent both modalities within =< 90 days of each other. The results are presented in table 6. The AUCs of the US parameters did not improve, ranging from 0.46 – 0.56, with all having 95% confidence intervals crossing 0.5. This suggests that the time window is not an effect modifier on the predictiveness of US parameters."

# Comment: 2) are there any previous studies that have similarly explored the predictive ability of Kidney US? The authors have not mentioed about them. It would be intersting to know what is the outcome in the previous studies.

# Response: There have been several studies which explored the predictiveness of US on predicting the “obstruction pattern” on DRS (reference numbers 11 - 15). However, there have not been studies that explored the predictiveness of US in predicting “non-diagnostic” DRS. We did additional searching and the closest we could find is a study by Jacobson et al. (reference 10) which analyzed the correlation between US and DRS, but did not analyze the ROC. We added this reference and added this discussion point.

# Revised text:

"A previous study by Jacobson et al. found a poor correlation between DRS and US findings in children with hydronephrosis with a Spearman correlation coefficient of only 0.24 [10]. However, to our knowledge, this current study is the first to examine the predictiveness of US parameters in relation to diagnostic vs. non-diagnostic DRS results."

# Comment: 3) Authors can mention any limiations in the current study. Also, mention the future directions for the researchers reading the manuscript.

# Response: We added a paragraph about the limitations and future directions at the end of the discussion section.

# Revised text:

"The main limitation of this study is its retrospective nature. Although all US studies were done by pediatric radiologists, since US is a highly operator-dependent modality, variations at the time the US studies were done are difficult to avoid. To address this issue, in this study, all US parameters were remeasured by one experienced pediatric radiologist (N.T.). Future prospective studies should aim to integrate multiple clinical, functional, and anatomical parameters to enhance prediction accuracy and improve patient selection for DRS."

# Comment: While revising your submission, please upload your figure files to the Preflight Analysis and Conversion Engine (PACE) digital diagnostic tool

# Response: I uploaded the figure file to PACE as instructed, and downloaded the PACE-adjusted version, and reuploaded them to the Editorial Manager submission system.

---

## [Decision Letter · Decision Letter 1]

25 Apr 2025

Can ultrasound parameters predict the diagnostic yield of diuretic renal scintigraphy in pediatric hydronephrosis?

PONE-D-25-01165R1

Dear Dr. Theerakulpisut,

We’re pleased to inform you that your manuscript has been judged scientifically suitable for publication and will be formally accepted for publication once it meets all outstanding technical requirements.

Kind regards,

Ahmet Çağlar, M.D.

Academic Editor

PLOS ONE

Additional Editor Comments (optional):

Reviewers' comments:

Reviewer's Responses to Questions

**Comments to the Author**

1. If the authors have adequately addressed your comments raised in a previous round of review and you feel that this manuscript is now acceptable for publication, you may indicate that here to bypass the “Comments to the Author” section, enter your conflict of interest statement in the “Confidential to Editor” section, and submit your "Accept" recommendation.

Reviewer #2: All comments have been addressed

2. Is the manuscript technically sound, and do the data support the conclusions?

Reviewer #2: Yes

3. Has the statistical analysis been performed appropriately and rigorously? 

Reviewer #2: Yes

4. Have the authors made all data underlying the findings in their manuscript fully available?

Reviewer #2: Yes

5. Is the manuscript presented in an intelligible fashion and written in standard English?

Reviewer #2: Yes

6. Review Comments to the Author

Reviewer #2: The authors have appropriately addressed all the comments. The manuscript is suitable for publishing.

7. PLOS authors have the option to publish the peer review history of their article (what does this mean? ). If published, this will include your full peer review and any attached files.

**Do you want your identity to be public for this peer review?** For information about this choice, including consent withdrawal, please see our Privacy Policy .

Reviewer #2: No

---

## [Editor Report · Acceptance letter]

PONE-D-25-01165R1

PLOS ONE

Dear Dr. Theerakulpisut,

I'm pleased to inform you that your manuscript has been deemed suitable for publication in PLOS ONE. Congratulations! Your manuscript is now being handed over to our production team.

Kind regards,

on behalf of

Dr. Ahmet Çağlar

Academic Editor

PLOS ONE